# Nine-Year Trends in Atrial Fibrillation Prevalence among Romanian Adult Hypertensives: A Post-Hoc Analysis of SEPHAR II-IV Surveys

**DOI:** 10.3390/ijerph19159250

**Published:** 2022-07-28

**Authors:** Cosmin Cojocaru, Aura-Elena Vîjîiac, Oana Gheorghe-Fronea, Teodora Mohaiu, Lucian Itu, Maria Dorobanțu

**Affiliations:** 1Faculty of Medicine, “Carol Davila” University of Medicine and Pharmacy, 020021 Bucharest, Romania; cosmin.cojocaru@drd.umfcd.ro (C.C.); aura.vijiiac@drd.umfcd.ro (A.-E.V.); oana.fronea@umfcd.ro (O.G.-F.); 2Cardiology Department, Emergency Clinical Hospital Bucharest, 014461 Bucharest, Romania; 3Faculty of Automatic Control and Computer Science, Polytechnic University of Bucharest, 060042 Bucharest, Romania; antonia.mohaiu.ext@siemens.com; 4Advanta, Siemens SRL, 500097 Brasov, Romania; lucian.itu@siemens.com; 5Department of Automation and Information Technology, Transilvania University of Brasov, 500174 Brasov, Romania; 6Romanian Academy, 010071 Bucharest, Romania

**Keywords:** atrial fibrillation, prevalence, arterial hypertension, epidemiological study, oral anticoagulation

## Abstract

Objectives: There are limited epidemiological data regarding atrial fibrillation (AF) in hypertensive (HT) Romanian adults. We sought to evaluate AF prevalence trends in the SEPHAR surveys (Study for Evaluation of Prevalence of Hypertension and Cardiovascular Risk in an Adult Population in Romania) during a nine-year interval (2012–2016–2021). Methods: Three consecutive editions of a national epidemiological survey regarding HT included representative samples of subjects stratified by age, gender and area of residence (SEPHAR II-IV—in total, 5422 subjects, mean age 48.69 ± 16.65 years, 57.5% (*n* = 3116) females). A post-hoc analysis of AF prevalence and oral anticoagulation (OAC) rates was performed. AF definition was based on a documented medical history of AF and/or AF documentation by study electrocardiogram. Results: General AF prevalence was 5.5% (*n* = 297). AF prevalence in HT subjects was 8.9% (*n* = 209) and has risen since SEPHAR II—7.2% (*n* = 57) and SEPHAR III—8.1% (*n* = 72) to SEPHAR IV—11.8% (*n* = 80), respectively (*p* = 0.001). AF prevalence has increased in HT males (SEPHAR II—5.3% (*n* = 19), SEPHAR III—7.6% (*n* = 26) and SEPHAR IV—11.7% (*n* = 35) (*p* = 0.010)) and in HT from urban areas (SEPHAR II—7.8% (*n* = 37), SEPHAR III—7.8% (*n* = 40), SEPHAR IV—14.7% (*n* = 50), *p* < 0.001). In SEPHAR III-IV, only 19.3% (*n* = 23) of HT AF patients with OAC indication were anticoagulated. Conclusions: AF prevalence has increased by ~64% in hypertensive Romanian adults between 2012 and 2021. However, anticoagulation strategies may be suboptimal in patients with cardioembolic risk.

## 1. Introduction

Hypertension (HT) is the main contributor to the global atrial fibrillation (AF) burden and it portends an excess AF risk of 50% in males and 40% in females [1]. All landmark AF trials (ROCKET-AF, RE-LY, ARISTOTLE and AUGUSTUS) have shown high prevalence of hypertension (>80–90%) in AF patients [2,3,4,5]. Furthermore, the overlap of HT and AF displays a J-curve effect regarding the incremental risk of both cardioembolic events and hemorrhagic complications at extreme blood pressure values [6,7].

Romania is characterized by very high cardiovascular morbidity and mortality rates. However, there are currently limited data regarding the overlap of AF and hypertension in Romanian adults, which mainly stem from EORP-AF/BALKAN-AF studies populations. Adherence to guideline-directed anticoagulation strategies appears to be insufficient both in terms of the lack of prescription of oral anticoagulants (OAC) in patients with embolic risk and the suboptimal time in therapeutic range (TTR), even when managed by cardiologists [8].

Consequently, we sought to evaluate the prevalence trends of AF and AF-inducing risk factors in hypertensive subjects included in the SEPHAR II (2012), III (2016) and IV (2021) (Study for Evaluation of Prevalence of Hypertension and Cardiovascular Risk in an Adult Population in Romania) cross-sectional national epidemiological studies.

## 2. Materials and Methods

### 2.1. Sample Selection and Data Collection in SEPHAR Surveys

The subject selection methodology in the SEPHAR II-IV surveys was based on a multistratified sampling procedure of Romanian adults aged between 18 and 80 years, which were randomly selected from the database of the Romanian population general direction of data records following the principle of equality of chances of being enrolled in the study. The recruitment criteria respected the population distribution regarding territorial regions (based on the recommendations of the National Institute of Statistics), type of residence (rural and urban), gender (men and women) and age groups (18–24, 25–34, 35–44, 45–54, 55–64 and 65–80 years) using the data from the last national census available. For an adult Romanian population of 16,269,839 adult citizens, of which 40.41% are estimated to be hypertensive patients based on SEPHAR II results [9], with a maximum error of ±2.18% at a confidence level of 95%, the minimum required sample size was 1379 study participants.

One month prior to the study conduction in each center, the selected study participants were informed about the survey conduction and their selection because of their demographic characteristics and were invited to send a response letter to the study organizers regarding their availability to participate in the study. Identification of the selected study participants respected the law for the protection of personal data of individuals, in such a manner that we did not contact a person with a precise identity but only a person with certain demographic characteristics (a person of a certain sex, of an age within a certain age category from a certain locality). *(The selection process and methodology have been rigorously detailed in previous publications* [9,10]). After confirming through a response letter, the subject was subsequently scheduled for an extensive evaluation in a mobile medical caravan at the nearest of the ten dedicated study centers in Romania in both rural and urban areas (Southern Romania—Bucharest, Pitești, Craiova; Eastern Romania—Iași, Constanța; Western and Central/Northern Romania—Timișoara, Arad, Cluj-Napoca, Oradea and Târgu-Mureș). The evaluation protocol consisted of two distinct visits separated by a four-day interval. The workflow is summarized in Figure 1.

Hypertension was defined as SBP at least 140 mmHg and/or DBP at least 90 mmHg at both study visits, using the arithmetic mean of the second and third BP measurement of each study visit (without taking into consideration the first BP measurement from either visit), or previously diagnosed hypertension under treatment during the previous 2 weeks, regardless of BP values. At each study visit, three BP measurements were performed at 1-min intervals using an automatic BP measuring device certified by the Association for the Advancement of Medical Instrumentation, ESH, and the British Society of Hypertension—model OMRON M6—with an adjustable cuff for arm circumferences from 24 to 42 cm, respecting the current guideline recommendations of the ESH and International Society of Hypertension [11].

Atrial fibrillation diagnosis was either based on EKG documentation of AF during the study visit or by a patient-disclosed history of diagnosed AF, irrespective of paroxysmal, persistent, long-standing or permanent type.

### 2.2. Statistical Analysis

Continuous data were expressed as mean ± standard deviation (SD) for normally distributed data and median (IQR; 25–75%) for non-normally distributed data. Categorical data were expressed as percentages (count). The normality of data was evaluated by the Kolmogorov–Smirnov test. Categorical variables were compared using Fisher’s exact test/chi-square analysis and continuous variables were compared using the Student *t*-test if normally distributed and non-parametric tests (Mann–Whitney U Test). Multivariable logistic regression was performed to assess the prognostic power of previously validated event predictors. Simple exponential smoothing by the additive damped trend method was used for the prediction of prospective AF prevalence based on previous prevalence values. A 2-sided *p*-value < 0.05 was considered statistically significant. Statistical analysis was performed using SPSS version 23 (IBM Corp., Armonk, NY, USA) software and Prism 9 (GraphPad Software, Graphpad Holdings, LLC, San Diego, CA, USA).

## 3. Results

### 3.1. AF Prevalence in Romanian Hypertensive Adults and Future Projections

#### 3.1.1. Overall AF Prevalence in SEPHAR Survey Global Population

A total of 5422 subjects were included in the SEPHAR II, III and IV surveys (42.5% (*n* = 2306) males and 57.5% (*n* = 3116) females), with a mean age of 48.69 ± 16.65 years. There were 43.7% (*n* = 2367) hypertensive subjects in the selected population. 

The overall prevalence of AF in all subjects enrolled in the SEPHAR surveys was 5.5% (*n* = 297). AF prevalence was higher in hypertensive compared to normotensive subjects (8.9% (*n* = 209) vs. 2.9% (*n* = 88), *p* < 0.001). Previously known hypertensives had a trend of higher AF prevalence compared to newly diagnosed hypertensives, yet not statistically significant (9.5% (*n* = 144) vs. 7.7% (*n* = 65), *p* = 0.132). AF prevalence was lower in controlled hypertensives compared to uncontrolled hypertensives, yet without statistical significance (8.5% (*n* = 134) vs. 11.1% (*n* = 73), *p* = 0.66). The summary of AF prevalence in specific subgroups in all subjects included in SEPHAR surveys is displayed in Table 1.

There was no overall gender-related difference in AF prevalence (5.7% (*n* = 177) in females vs. 5.2% (*n* = 120) in males, *p* = 0.469). AF prevalence had a higher trend in hypertensive females compared to hypertensive males, however without statistical significance (9.5% (*n* = 129) vs. 8% (*n* = 80), *p* = 0.213). AF prevalence was higher in progressively older age subgroups (13.9% (*n* = 159) in subjects aged over 60 years, 4.3% (*n* = 41) in subjects aged 40–59 years, 3.4% (*n* = 9) in subjects younger than 39 years, *p* < 0.001).

Similarly, there was no overall difference in AF prevalence between urban and rural areas of residence (5.8% (*n* = 182) in urban areas vs. 5.1% (*n* = 114) in rural areas, *p* = 0.303), nor specifically for hypertensive subjects (9.6% (*n* = 127) in urban areas vs. 7.9% (*n* = 81) in rural areas, *p* = 0.164). T2DM hypertensive subjects had a higher AF prevalence than non-diabetic hypertensive subjects (14.8% (*n* = 68) vs. 7.1% (*n* = 116), *p* < 0.001). Obese hypertensive subjects had a higher AF prevalence than non-obese hypertensive subjects (10.1% (*n* = 143) vs. 6.9% (*n* = 64), *p* = 0.004).

AF diagnosis was obtained by EKG and by the presence of a history of AF in 1.7% (*n* = 87) and 4.5% (*n* = 238) of patients, respectively. AF diagnosis by EKG was more frequent in hypertensive subjects than normotensive subjects (2.9% (*n* = 65) vs. 0.9% (*n* = 22), *p* < 0.001). Hypertensive subjects had a history of AF more frequently than normotensives (7.3% (*n* = 170) vs. 2.3% (*n* = 68), *p* < 0.001).

#### 3.1.2. Trends of AF Prevalence from 2012 (SEPHAR II) to 2021 (SEPHAR IV)

The overall prevalence of AF has progressively risen in SEPHAR II, III and IV in the overall study population, irrespective of hypertensive status (3.8% (*n* = 75), 5.4% (*n* = 107) and 7.8% (*n* = 115), respectively, *p* < 0.001). In particular, AF prevalence has risen in hypertensive subjects (7.2% (*n* = 57), 8.1% (*n* = 72) and 11.8% (*n* = 80), respectively, *p* = 0.001) (Figure 2).

AF prevalence has significantly risen in hypertensive males over the three evaluated studies: SEPHAR II—5.3% (*n* = 19), SEPHAR III—7.6% (*n* = 26) and SEPHAR IV—11.7% (*n* = 35) (*p* = 0.010). In contrast, AF prevalence in hypertensive females has not significantly increased (SEPHAR II—8.8% (*n* = 38), SEPHAR III—8.4% (*n* = 46), SEPHAR IV—11.9% (*n* = 45), *p* = 0.173).

A significant increase in AF prevalence has been observed in urban hypertensive subjects (SEPHAR II—7.8% (*n* = 37), SEPHAR III—7.8% (*n* = 40), SEPHAR IV—14.7% (*n* = 50), *p* < 0.001). However, the AF prevalence in rural hypertensives has not risen significantly (SEPHAR II—7.2% (*n* = 57), SEPHAR III—8.1% (*n* = 72), SEPHAR IV—11.7% (*n* = 79), *p* = 0.424).

The highest AF prevalence was observed in hypertensive subjects aged 60 years or older in SEPHAR IV (16.3% (*n* = 63) in comparison with hypertensives aged 40–59 years (5.7% (*n* = 14)) and hypertensives younger than 39 years (6.3% (*n* = 3), *p* < 0.001). Furthermore, AF prevalence in SEPHAR IV was three-fold higher in hypertensive subjects aged 60 years or more compared to normotensive subjects from the same age category (16.3% (*n* = 63) vs. 4.4% (*n* = 35), *p* < 0.001). In particular, in hypertensive subjects aged 60 years or older, AF prevalence was highest in SEPHAR IV (16.3% (*n* = 63)) compared to SEPHAR III (13.1% (*n* = 55) and SEPHAR II (12% (*n* = 41); however, it did not reach statistical significance (*p* = 0.211).

The forecasted AF prevalence in 2028 may increase up to 14.7% (Figure 3) as evaluated by exponential smoothing analysis based strictly on previously observed prevalence in SEPHAR surveys.

### 3.2. Global Epidemiology and Trends of AF-Inducing Risk Factor Epidemiology in SEPHAR Surveys

The global prevalence values or mean value ± SD of AF-inducing risk factors in hypertensive subjects with AF and hypertensives without AF are summarized in Table 2.

The prevalence trends of AF-inducing risk factors in the SEPHAR II, III and IV surveys are presented in Table 3.

The prevalence of hypertension has risen in males included in SEPHAR II (38.4% (*n* = 360)) and SEPHAR III (43.9% (*n* = 341)) up to SEPHAR IV (50.8% (*n* = 300), *p* < 0.001). In contrast, hypertension prevalence in females has not changed significantly over the three evaluated studies (SEPHAR II—42.2% (*n* = 438), SEPHAR III—45.9% (*n* = 548) and SEPHAR IV—42.8% (*n* = 379), *p* = 0.174). AF patients demonstrated higher blood pressure (BP) values than non-AF subjects (systolic BP—137.36 ± 20.77 vs. 131.02 ± 19.83 mmHg, *p* < 0.001 and diastolic BP—84.03 ± 10.70 vs. 81.51 ± 10.88, *p* < 0.001).

The SEPHAR survey edition (SEPHAR IV versus previous editions) independently predicted AF in hypertensive patients after adjustment for age (OR 1.413 (CI 95% 1.046–1.910, *p* = 0.024)). In our multivariable regression model (Figure 4), age (OR 1.044, CI 95% 1.028–1.060), survey edition (OR 1.504, CI 95% 1.022–2.213), T2DM (OR 1.450, CI 95% 1.000–2.102) and previously diagnosed HF (OR 5.386, CI 95% 3.642–7.966) independently predicted AF in hypertensive patients. Imaging-derived AF predictors were excluded from the model due to the lack of data availability in the SEPHAR II survey.

### 3.3. Rates of Oral Anticoagulation (OAC) in AF Patients in SEPHAR III and IV Surveys

Figure 5 summarizes the results of a pooled analysis of SEPHAR III and SEPHAR IV AF patients regarding the distribution of CHA_2_DS_2_-VASc scores. The mean CHA_2_DS_2_-VASc value in SEPHAR III and IV was 3.00 ± 1.76 points in AF patients, whereas in AF hypertensive patients, it was 3.71 ± 1.45 points (versus AF normotensive subjects, which scored 1.47 ± 1.33 points, *p* < 0.001). Based on CHA_2_DS_2_-VASc scores, 81.5% (*n* = 181) of subjects and all of the AF hypertensive subjects had OAC indication as recommended by the guidelines. Only 17.9% (*n* = 26) of AF patients (and particularly only 19.3% (*n* = 23) of hypertensive AF patients) with OAC indication were receiving treatment with OAC. AF patients in SEPHAR IV were receiving Apixaban in 33.4% (*n* = 9), Rivaroxaban in 29.6% (*n* = 8), vitamin K antagonist (VKA) in 25.9% (*n* = 7) and Dabigatran in 11.1% (*n* = 3) of cases, respectively, as OAC therapy. Data regarding specific OAC type were unavailable for analysis for SEPHAR II and III.

Patients with OAC indication had a lower income (1937.60 ± 2010.384 vs. 2432.62 ± 2521. 292 lei, *p* < 0.001) and were more frequently from rural areas compared to those without an OAC indication (OAC indicated—rural 43.7% vs. urban 56.3%; without OAC indication—rural 39.7% vs. urban 60.3%, *p* = 0.003). Patients with OAC indication had less frequently graduated from university studies and more frequently high school and primary studies compared to those without OAC indication (for those with OAC indication: university studies 8.9% (*n* = 210), high school studies 46% (*n* = 1080), primary school 37.3% (*n* = 875) and no formal education 7.8% (*n* = 182) versus those without OAC indication: university studies 18.5% (*n* = 377), high school studies 41.3% (*n* = 843), primary school 33.4% (*n* = 682) and no formal education 6.8% (*n* = 138), *p* < 0.001).

The general prevalence of ischemic stroke history (irrespective of mechanism) in SEPHAR III and IV was 3.3% (*n* = 114). Ischemic stroke was more prevalent in hypertensives at 5.9% (*n* = 89) versus normotensives at 1.4% (*n* = 25) (*p* < 0.001) and in AF patients (8.7% (*n* = 19) versus 3.1% (*n* = 95), *p* < 0.001). Out of the nineteen AF patients with ischemic stroke, fifteen (78.94%) were not receiving OAC.

## 4. Discussion

### 4.1. AF Prevalence in Romania Appears to Be Higher than the Average European Prevalence and Has Been Increasing during the Last Decade

Although heterogenous, the most recent epidemiological data have underlined two central features regarding global AF prevalence in adults: it is estimated to vary between 2% and 6% and is even higher than 10% in elders and in the presence of structural heart disease [12,13,14,15,16]. AF may affect more than one fifth of patients aged 85 years or older [17]. It is highly prevalent in hypertensive patients, as hypertension raises the risk of incident AF by 50% [18]. Conversely, it is known from landmark AF trials that AF patients are frequently (>80% of cases) hypertensive [2,3,4,5]. Approximately 74.1% of Romanian AF patients included in the BALKAN-AF survey were hypertensives [8].

The observed global AF prevalence in the SEPHAR surveys (5.5%), and, particularly, AF prevalence in hypertensives (8.9%), is strikingly higher than previously published data. For instance, most European studies have reported AF prevalence rates between 7.5 and 9% in subjects aged 60 years or older [17,19,20,21,22], which is lower than our reported subgroup AF prevalence (13.9%). Asian studies have reported AF prevalence rates of 3.46% in hypertensive subjects [23].

Most importantly, AF prevalence has shown incremental changes during the last decade and is estimated to double by 2060 [17,22]. AF prevalence in hypertensives has increased ≅64% since 2012 up to 2021 from 7.2% to 11.8% in our particular dataset, especially driven by AF in males residing in urban areas. In this sense, public health policy changes are mandated: firstly, for improving the identification of such cluster profiles at risk for AF, especially those highly susceptible to stroke; secondly, to formulate community-based interventions to prevent AF and to correctly implement treatment if already diagnosed.

We hypothesized that the growing prevalence of AF is attributable to the lack of control of cardiovascular and AF-inducing risk factors in an ageing population.

Firstly, age independently predicted AF in our dataset and has shown a significant increase during the SEPHAR surveys. It is already known that a significant increase of 3% in the share of the population aged 65 years or older in Romania has taken place between 2012 and 2021, which is consistent with other European Union countries and reflects the process of demographic ageing [24]. In this sense, the BALKAN-AF Romanian subgroup (which recruited consecutive AF patients) was the oldest (70.9 ± 10.8 years old) compared to the other participating countries [8]. Advanced age is the most prominent risk factor for AF and, most importantly, an independent risk factor for stroke [14].

Secondly, our data show the growing prevalence of typical AF-inducing risk factors such as hypertension, T2DM, obesity and OSA, which may account for the rising AF presence in Romanians. The relationship between AF, hypertension and T2DM is well documented and is mediated by the development of atrial cardiomyopathy and multiple other complex mechanisms. Additionally, the risk of AF was increased by up to 50% by hypertension in the Framingham study [1]. In the ARIC (Atherosclerosis Risk in Communities) study, 20% of incident AF cases were attributed to hypertension [25]. Even if there still is paucity in published data, the Cardio-Sis trial has provided insight into how adequate BP control can be protective from new-onset AF in hypertensives [26]. Both obesity (as defined by BMI > 30 kg/sqm) and visceral obesity (waist circumference > 94 cm in males and 80 cm in females) have become more prevalent in Romania and are well-known drivers of AF epidemiology. Meta-analyses have shown that a 5% weight gain raises the risk of incident AF by 13% [27], while a > 10% weight loss significantly reduces AF recurrences [28]. Our proposed multivariable prediction model suggests that HF, age and T2DM are the main drivers of AF prevalence in hypertensive Romanians. However, considering that the survey edition itself independently predicted AF in hypertensives, it may be inferred that other risk factors that could not currently be accounted for in this post-hoc analysis are becoming progressively more prevalent in Romania and act as significant promoters of AF in this high cardiovascular risk population.

In conclusion, intensive control and careful identification of modifiable AF-inducing risk factors are crucial for lowering prospective AF prevalence in Romania.

### 4.2. Oral Anticoagulants Are Severely Underutilized Even in High Cardioembolic Risk Subjects

Our data provide insight into the potential underutilization of OACs in Romanian AF subjects with formal guideline recommendations [12]. In our particular dataset, less than 20% of patients were receiving OACs at the time of the SEPHAR III and IV surveys, whereas the mean CHA_2_DS_2_-VASc score in hypertensive AF patients was 3.71 ± 1.45 points. This, however, significantly contrasts both the previously reported OAC rates in the BALKAN-AF survey (73.6% in Romania) [8] and in large-scale European registries (EORP-AF Pilot, EORP-AF General Long-Term and PREFER-AF) of more than 80% in subjects with cardioembolic risk (i.e., CHA_2_DS_2_-VASc ≥ 2 in females and ≥ 1 in males) [22,29,30,31]. This discrepancy may partially result from the enrolment methods. All subjects included in the aforementioned studies had been diagnosed with AF and were recruited from cardiology and internal medicine practices (both outpatient and inpatient), as opposed to the SEPHAR surveys, in which the targeted population was a result of multistratified sampling, irrespective of prior specialist management. This may lead to significant differences in treatment quality with regard to formal guideline indications.

Furthermore, these hypotheses regarding OAC prescription result from the analysis of a relatively limited population of 222 AF patients from SEPHAR III and IV (which were not designed to assess AF prevalence and the quality of its medical treatment). Consequently, for the adequate evaluation of AF epidemiology in Romania, selection of the study population should be tailored specifically to AF projected epidemiology and not hypertension (as in the SEPHAR surveys).

Therefore, these observations should be scrutinized as they may raise awareness regarding suboptimal AF treatment in primary care or in subjects with limited contact with medical facilities, which can impact subsequent clinical outcomes.

Hypertension is the most important epidemiological target for stroke prevention as it is associated with the highest population-attributable risk (PAR) for ischemic stroke (40.7–54.8%) [32]. Even if it exerting less impact, AF-related PAF for ischemic stroke has been estimated at 3.1% in South Asia and reaches ≅17% in Europe, North America and Australia [32]. The SPORTIF trials demonstrated incremental risks of stroke in AF patients with systolic BP > 140 mmHg, which shows the additive effect of HT-AF overlap [33]. We observed that a higher than two-fold stroke prevalence was observed in AF patients compared to those without AF (mainly driven by those without OAC).

Patients with an OAC indication were more frequently from rural areas, had lower income and had received less formal education. Furthermore, only 72.1% of patients from the SEPHAR IV survey were adherent to treatment based on the four-item Morisky Medication Adherence Scale. These may be potential factors for OAC underutilization due to difficult access to medical services or prescription medicine. In this sense, direct oral anticoagulants (DOACs) are not fully reimbursed by medical insurance in Romania.

Even so, in those receiving OAC, there was a higher rate of prescription of DOACs in comparison to vitamin K inhibitors, which is consistent with existing evidence of higher DOAC prescription and the superior cost-effectiveness of DOACs in real-world practice [34,35]. This may provide insight regarding the progressive shift from the predominantly VKA-based OAC therapy (69% in 2016 [8]) to DOACs in Romania, despite suboptimal rates of OAC prescription.

## 5. Limitations

We have emphasized that this is a post-hoc analysis of a cross-sectional survey specifically dedicated to arterial hypertension epidemiological data (and not AF prevalence) in Romanian adults. AF diagnosis was only based on EKG documentation and AF patient history. In this sense, an AF history-based diagnosis is subject to the significant variability of AF-screening programs in various areas of residence. Furthermore, the lack of Holter-based monitoring may lead to underdiagnosis of paroxysmal AF episodes.

Anticoagulation rates were reported as a sub-analysis of SEPHAR III and IV populations (anticoagulation data were unavailable in SEPHAR II). Subjects were included in this sub-analysis irrespective of prior dedicated specialized management, which may impact anticoagulation rates (in contrast to the previously reported results from European and/or Balkan AF-dedicated registries, in which the majority of subjects were enrolled by cardiologists).

## 6. Conclusions

Atrial fibrillation prevalence has increased by approximately 64% in hypertensive Romanian adults between 2012 and 2021. However, oral anticoagulation may be underutilized in real-world AF management in Romania.

## Figures and Tables

**Figure 1 ijerph-19-09250-f001:**
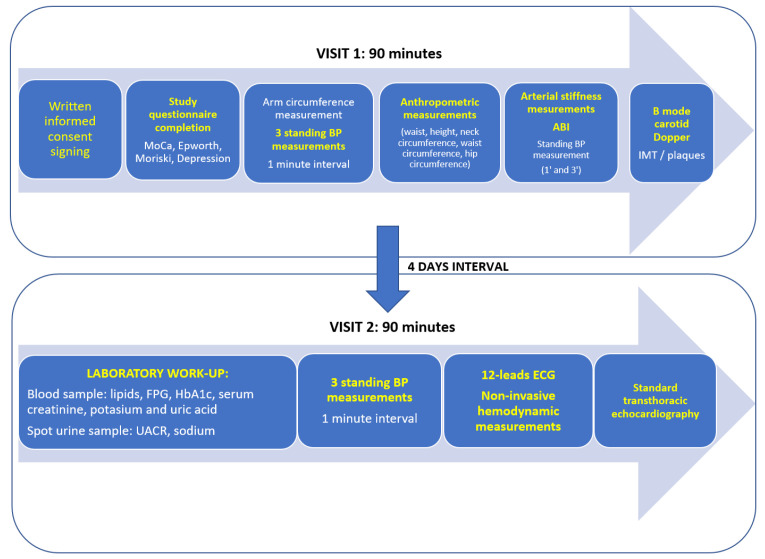
Study evaluation workflow in SEPHAR surveys. BP = blood pressure, ABI = ankle-brachial index, IMT = intima-media thickness, FPG = fasting plasma glucose, UACR = urinary albumin/creatinine ratio, ECG = electrocardiogram, MoCA = Montreal Cognitive Assessment, ECG = electrocardiogram.

**Figure 2 ijerph-19-09250-f002:**
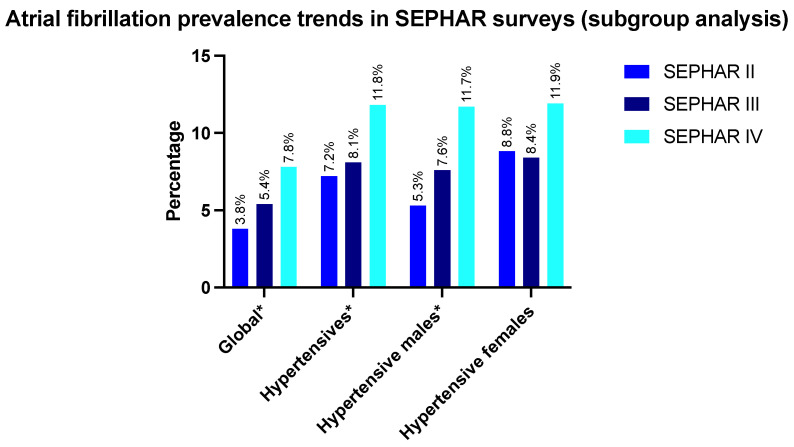
Atrial fibrillation prevalence trends in hypertensive subjects enrolled in SEPHAR II, III and IV surveys in Romania, * *p* < 0.05 for inter-group comparison.

**Figure 3 ijerph-19-09250-f003:**
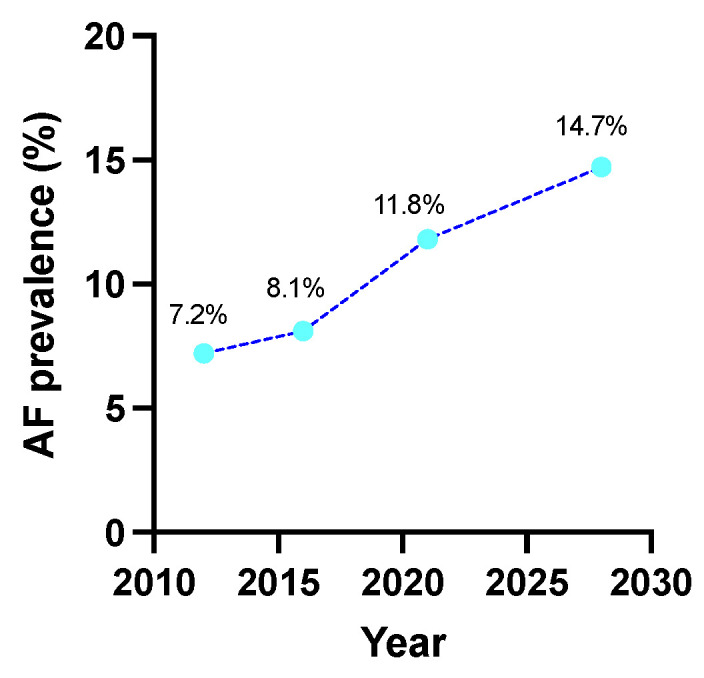
Exponential smoothing analysis by additive damped trend method for forecasted prevalence of AF in hypertensive Romanian adults. AF = atrial fibrillation.

**Figure 4 ijerph-19-09250-f004:**
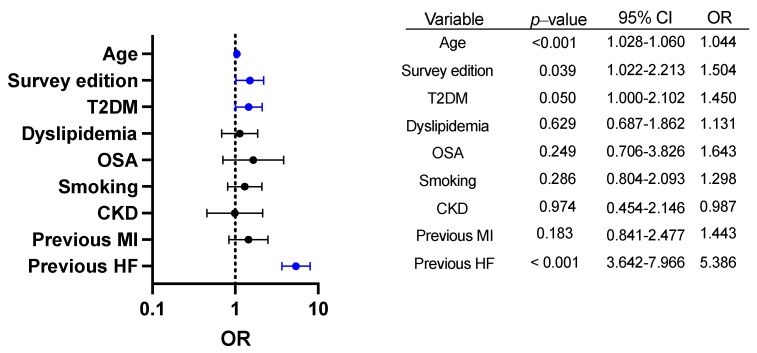
Multivariable logistic regression model including typical AF-inducing risk factors in hypertensive patients. Independent predictors in multivariable logistic regression analysis for AF status (age, survey edition, previous HF, T2DM) are highlighted in blue. Error bars denote 95% confidence intervals for odds ratio (OR). T2DM = type 2 diabetes mellitus, OSA = obstructive sleep apnea, CKD = chronic kidney disease, MI = myocardial infarction, HF = heart failure, AF = atrial fibrillation.

**Figure 5 ijerph-19-09250-f005:**
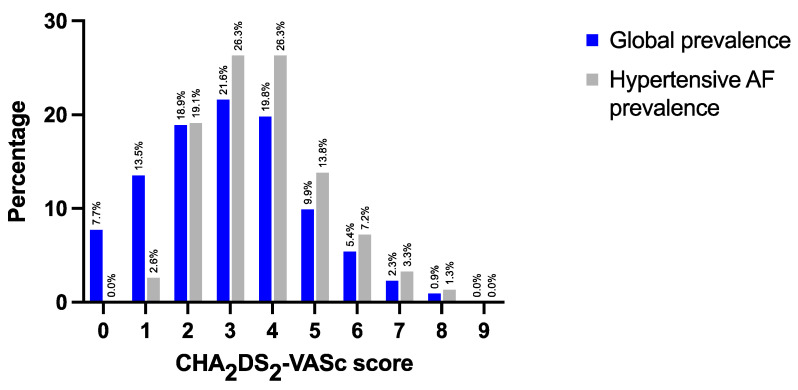
CHA_2_DS_2_-VASc score values in AF patients from SEPHAR III and SEPHAR IV surveys. SEPHAR = Study for Evaluation of Prevalence of Hypertension and Cardiovascular Risk in an Adult Population in Romania.

**Table 1 ijerph-19-09250-t001:** Overall and subgroup atrial fibrillation prevalence in SEPHAR II, III and IV surveys in the presence and the absence of hypertension, respectively. * = *p*-value for prevalence comparison in hypertensive versus normotensive subjects, T2DM = type 2 diabetes mellitus, BMI = body mass index.

	All Subjects, *n* (%)	Hypertensive Subjects, *n* (%)	Normotensive Subjects, *n* (%)	*p*-Value *
Global population	297 (5.5)	209 (8.9)	88 (2.9)	<0.001
Males	120 (5.2)	80 (8)	40 (3.1)	<0.001
Females	177 (5.7)	129 (9.5)	48 (2.7)	<0.001
<40 years	25 (1.4)	9 (3.4)	16 (1.1)	0.008
40–60 years	75 (3.6)	41 (4.3)	34 (3)	0.126
>60 years	197 (12.6)	159 (13.9)	38 (9)	0.01
Urban residence	182 (5.8)	127 (9.6)	55 (3)	<0.001
Rural residence	114 (5.1)	81 (7.9)	33 (2.7)	<0.001
T2DM	74 (11.8)	68 (14.8)	6 (3.6)	<0.001
Obesity	192 (7.2)	143 (10.1)	49 (3.9)	<0.001

**Table 2 ijerph-19-09250-t002:** AF-inducing risk factors in hypertensive subjects with atrial fibrillation and hypertensive subjects without atrial fibrillation. T2DM = type 2 diabetes mellitus, OSA = obstructive sleep apnea, HF = heart failure, LA = left atrium, AF = atrial fibrillation, MI = myocardial infarction.

Risk Factor	Hypertensives with AF	Hypertensives without AF	*p*-Value
Male gender, *n* (%)	80 (38.3)	921 (42.8)	0.213
Age, mean ± SD	66.14 ± 12.23	56.76 ± 14.30	<0.001
Urban residence, *n* (%)	127 (61.1)	1201 (55.9)	0.164
Dyslipidemia, *n* (%)	173 (84)	1749 (82.1)	0.567
Active smoking, *n* (%)	30 (14.5)	414 (19.7)	0.079
Obesity, *n* (%)	108 (52.2)	993 (46.5)	0.126
T2DM, *n* (%)	68 (37)	392 (20.4)	<0.001
OSA history, *n* (%)	10 (4.9)	48 (2.3)	0.032
Hypertension control, *n* (%)	134 (64.7)	1438 (71)	0.066
HF diagnosis history, *n* (%)	84 (41.6)	166 (7.9)	<0.001
MI history, *n* (%)	27 (13.2)	96 (4.6)	<0.001
Systolic dysfunction, *n* (%)	9 (9.3)	47 (5.1)	0.100
Diastolic dysfunction, *n* (%)	64 (50.4)	570 (47.3)	0.515
LA indexed volume, mean ± SD	44.65 ± 25.42	29.52 ± 10.94	<0.001

**Table 3 ijerph-19-09250-t003:** Trends in AF-inducing risk factors in SEPHAR II, SEPHAR III and SEPHAR IV surveys. T2DM = type 2 diabetes mellitus, OSA = obstructive sleep apnea, HF = heart failure.

Risk Factor	SEPHAR II	SEPHAR III	SEPHAR IV	*p*-Value
Age	47.03 ± 15.57	48.54 ± 17.49	51.19 ± 16.61	<0.001
Hypertension, *n* (%)	798 (40.4)	889 (45.1)	680 (46)	0.001
Obesity, *n* (%)	535 (27.4)	684 (34.7)	579 (39.6)	<0.001
Visceral obesity, *n* (%)	1607 (81.7)	1462 (74.2)	1114 (76.2)	<0.001
Smoking, *n* (%)	532 (27.1)	470 (23.9)	383 (25.9)	0.059
T2DM, *n* (%)	201 (10.2)	240 (12.2)	186 (20.2)	<0.001
Dyslipidemia, *n* (%)	1438 (73)	1522 (77.3)	974 (68)	<0.001
OSA, *n* (%)	9 (0.5)	41 (2.2)	48 (3.2)	<0.001
History of diagnosed HF, *n* (%)	80 (4.2)	129 (6.5)	89 (6.1)	0.004
History of myocardial infarction, *n* (%)	44 (2.3)	70 (3.8)	42 (2.9)	0.03
Diastolic dysfunction, *n* (%)	*n*/A	827 (42)	359 (36.8)	0.007
Systolic dysfunction, *n* (%)	*n*/A	58 (3.7)	27 (3.8)	0.999

## Data Availability

Archived datasets are available upon request by any interested third party.

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
