# Peer review of "Nine-Year Trends in Atrial Fibrillation Prevalence among Romanian Adult Hypertensives: A Post-Hoc Analysis of SEPHAR II-IV Surveys"

_ijerph, 2022, doi:10.3390/ijerph19159250_

Round 1
Reviewer 1 Report
Congratulations on this interesting work. Romania is one of the regions in Europe with the highest cardiovascular risk. Given the scientific and technological evolution in the medical field, and that the population trend in Romania is a downward one, it would have been interesting to study thoroughly the reasons for the significant increase in the prevalence of atrial fibrillation.
Hypertension and afib are crucial public health problems and should be seen as priorities, mainly because they tend to trigger or enhance one another. This paper's central question addresses the need to know the prevalence of both htn and afib in the Romanian population and the anticoagulant treatment. The research of the current paper started from the need to search for more data regarding hypertension and atrial fibrillation co-existence in the Romanian population. Unfortunately, the existing data in this region is limited, and there is a need for switching the paradigm in this European area because it is on the map of very high cardiovascular risk, which means it is mandatory to find and repair the changeable causes for this unfortunate statistic. The authors performed a posthoc and exhaustive study on large samples extracting data from SEPHAR II-IV by researching the prevalence of atrial fibrillation through personal history or ECG.
The current paper is relevant because it proved that afib is more prevalent in htn pts versus non-htn pts. Also, The fact that this trend is ascending should represent an alarming finding for physicians and general politics and should lead to immediate changes regarding the approach of hypertensive patients with a more extensive search for afib. A better screening program and the thought of searching for afib in all htn pts could improve the numbers. Also, the increased prevalence in the male gender and urban areas should lead to further research for possible reasons and triggers in the lifestyle. Another critical finding with possibly fatal consequences is the under-treatment regarding anticoagulation in patients with a clear indication. Only 19.3% of HTN+AF patients who needed anticoagulation were prescribed the oral anticoagulant, meaning that almost 80% were not treated and were exposed to thromboembolic events. The most surprising finding is that the prevalence increased in just nine years by 64% and that less than a quarter of the pts with anticoagulation indications receive the correct treatment, which is a clear indicator of the need for changing the approach.
In the general cardiovascular field, the topic is not original. However, its originality arises from the country it was performed in, where the data is limited. It adds data from the Romanian area, where there is limited information and the numbers are not in favour of the patients, meaning that further research and public health measures need to be performed to improve the quality of life and survival.
The paper is written well, with minor language misspellings and typos, but the text is clear and accessible. Nonetheless, it would have been better to have a more detailed approach regarding the methodology instead of referring to the previous publications, although the same team performed them.
The conclusion was consistent with the evidence and arguments in the results section.
In conclusion, they did address the central question adequately, focusing on the prevalence of afib in htn pts and the oral anticoagulant prescription in pts with a clear indication for it.
Author Response
Dear Sir/Madam,
We respectfully thank you and highly appreciate your observations and your proposed revisions. We hereby provide a point-by-point response to your comments and the updated manuscript in accordance to all reviewers' observations. We hope to have improved the content of the manuscript text during the revision process.
We have attached the response as a "Word" document (Please see the attachment).
With high respect for the revision process,
The Authors

Reviewer 2 Report
Cojocaru et al. investigated the prevalence of atrial fibrillation and the frequency of OAC use using the Romanian hypertension database. Although it's very interesting, I don't think this paper reaches the level published by IJERPH.
Major points
1) Author mentioned that the prevalence of atrial fibrillation is increasing year by year in SEPHAR Survey II III IV, but there is a significant difference in age among II III IV. Authors already reported in this study that age is related to the onset of atrial fibrillation, so it should be corrected by age and statistics should be performed.
2) Since the frequency of use of OAC is 20% or less, please show the data such as the incidence of cerebral infarction.
3) Please clarify the frequency of the warfarin usage.
Minor point
1)Please revise the style and syntax etc.
For example, in line 260, there is a double space in front of CHAD2DS2-VASc.
For example, in line 235, "between 2021 and 2021”, isn’t it strange? I think it is "between 2012 and 2021".
Author Response

(The authors gave the same response as above.)

Reviewer 3 Report
interesting work, well designed, described and conclusion consistent with data.
please explain better in line 200 (rural vs rural).
Author Response

(The authors gave the same response as above.)

Round 2
Reviewer 2 Report
The response is courteous and appropriate.
One point, please check the following
“Line 242, Page 7, Results: “Figure 5 summarizes the results of a pooled analysis of SEPHAR III and SEPHAR IV AF patients regarding the distribution of CHA2DS2-VASc scores. The mean CHA2DS2-VASc value in SEPHAR III and IV was 3 ± 1.76 points in AF patients, whereas in AF hypertensive patients it was 3.71 ± 1.45 points (versus AF normotensive subjects which scored 1.47 ± 1.33 points, p < 0.001).”
3 ± 1.76 points should be 3.00 ± 1.76 points in terms of significant digits